# The influence of generative artificial intelligence usage on employees' innovative job performance

**Hui Zhang, Lidong Zhu, Ayuan Zhang****\*, Kilichov Shohruh**

School of Economics and Management, Anhui Normal University, Wuhu, Anhui, China

\* 18298257065@163.com

## Abstract

The rapid advancement of AI technology has accelerated the adoption of generative artificial intelligence (GenAI) tools in the workplace, eroding the boundaries between professional responsibilities and personal space, thus impacting employees' innovative performance. This study empirically examines the link between innovative job performance and GenAI tool usage, framed through the Uses and Gratifications Theory. Analyzing survey data from 366 employees nationwide revealed that: (1) both cognitive and social uses of GenAI tools significantly enhance innovative performance; (2) cognitive use primarily facilitates knowledge transfer behaviors, while social use bolsters resource acquisition. Enhanced knowledge transfer and resource acquisition, in turn, improve job satisfaction, which is pivotal in driving innovative performance. This study introduces a novel framework for utilizing GenAI tools to optimize and manage employee performance within organizational settings.

## 1. Introduction

Generative artificial intelligence (GenAI) refers to algorithms that autonomously generate original textual, visual, or audio content, mimicking human creativity and reasoning [1,2]. Its adoption in the workplace has surged as organizations leverage GenAI to boost productivity, streamline decision-making, and foster innovation. The 2024 Microsoft Work Trend Index reports that over 90% of employees in China utilize GenAI for daily tasks, with more than half integrating AI tools into core business operations [3]. This widespread adoption highlights GenAI's potential to drive innovation and maintain a competitive edge [4].

Innovative job performance (IJP) encompasses employees' proactive involvement in idea generation, problem-solving, and process improvement beyond routine tasks [5]. Recent research highlights the critical role of technological enablement and knowledge dynamics in shaping IJP within digital organizations [6,7]. However, the impact of GenAI on innovation outcomes remains debated—some scholars argue that over-reliance on algorithms may hinder independent thinking [8], while others suggest that responsible GenAI use fosters creativity and cross-disciplinary collaboration [9,10].

**Data availability statement:** All relevant data for this study are publicly available from the Zenodo repository (https://doi.org/10.5281/zenodo.18180981).

**Funding:** This research was funded by the National Natural Science Foundation of China (NSFC), grant number "72202002". The funders had no role in study design, data collection and analysis, decision to publish, or preparation of the manuscript.

**Competing interests:** The authors have declared that no competing interests exist.

Building on Uses and Gratifications Theory (UGT), this study explores how employees utilize GenAI to satisfy cognitive and social needs that influence innovation. Cognitive use refers to leveraging GenAI for gathering and processing work-related knowledge, while social use involves using AI tools for collaboration, communication, and professional networking—such as co-creating content with colleagues *via* ChatGPT or coordinating cross-departmental tasks using AI assistants [11]. Both types of use promote collective intelligence and encourage innovative behavior. Furthermore, this study refines the concept of resource acquisition, defined as employees' ability to access informational, material, and relational resources that support innovation. GenAI facilitates this access through intelligent search, recommendations, and network-based collaboration tools [12]. By integrating cognitive and social motivations with resource acquisition, this research offers a contemporary framework for understanding how GenAI adoption enhances employees' IJP in modern organizational environments.

## 2. Conceptual background

### 2.1. Uses and gratifications theory

Originating from mass communication research, UGT provides a structural framework for analyzing individuals' intentional media choices aimed at fulfilling psychological needs [13]. The theory's core premise posits that users engage with media deliberately, driven by specific needs, and seek personalized benefits from technological features [14,15]. In the context of employee behavior, the theory offers valuable insights into how employees utilize various tools and technologies to stimulate innovation [17,18]. For example, Zhang et al. found that both work-related and social media usage positively impacted employees' innovative performance, surpassing conventional performance [18].

### 2.2. Classification of motivations for the use of GenAI

Most previous research on GenAI has approached user behavior toward these tools as a uniform, monolithic pattern. However, as technology evolves—especially tools like GenAI that support diverse applications—this broad perspective overlooks the complexity and multidimensional nature of user behavior. Consequently, a more nuanced approach is necessary to examine the diverse patterns of GenAI usage and the mechanisms influencing them across varying contexts [16,19–22]. Drawing on UGT, this framework links users' behavioral choices to their outcomes, making it well-suited for exploring the motivations behind GenAI adoption and its influence mechanisms in the workplace [23,24]. Accordingly, this paper aims to systematically categorize and analyze the specific behavioral patterns associated with employees' use of GenAI tools in work settings, with a focus on their intrinsic motivations for adopting these technologies.

The original UGT provides a foundation for understanding media engagement motivations, typically categorized into three primary dimensions: social, hedonic, and cognitive motivations [17]. Social motivation involves using media to connect with others who share similar interests or topics [25]. Hedonic motivation is driven by

the desire to maximize leisure through enjoyable media experiences, such as optimizing relaxation [26], reducing occupational stress [25], and engaging in recreational rejuvenation [27]. Cognitive motivation encompasses two key aspects: collaborative knowledge development, often facilitated by digital platforms like WeChat [27,28], and using media for learning and development. According to Skjuve, individuals primarily utilize GenAI tools to enhance productivity, stimulate creativity, and engage in social interaction, learning, and development [29]. Some scholars argue that social and cognitive motivations are particularly relevant in the work context. For instance, Zhang distinguished between social and task-related motivations for social media use [18]. While social media primarily impacts personal interactions, GenAI's role in the workplace is more closely linked to knowledge management [30], innovation stimulation, remote work support [25], and social collaboration [26,28,30]. In contrast, GenAI does not significantly influence employees' relaxation or time-killing behaviors. Building on these insights, this study posits that employees' motives for using GenAI tools in the workplace can be categorized into "cognitive" and "social" motives. Accordingly, the study suggests that employees' behaviors in using GenAI tools can be classified into two distinct categories: cognitive-based and social-based GenAI use behaviors. Cognitive-based behaviors involve sharing relevant knowledge and accessing or interpreting information through GenAI tools [27,28], while social-based behaviors involve using GenAI tools to find like-minded partners, establish new social relationships, and expand professional networks [25].

## 3. Research hypothesis

### 3.1. Cognitive use of GenAI tools and knowledge transfer

Knowledge transfer refers to the deliberate exchange of both explicit and tacit knowledge between different organizational units, facilitated through various communication channels [31]. Knowledge management frameworks suggest that cognitive GenAI adoption accelerates knowledge acquisition in two main ways: by expediting transactional processes and enhancing cognitive absorption [32]. GenAI systems optimize the management of informational repositories through algorithms that filter, semantically summarize, and retrieve content contextually [33]. For example, Morgan Stanley utilized GPT-4 to process 100,000 company-specific investment documents, refining their knowledge assistance system for financial advisors [33]. GenAI-powered digital interfaces transform complex information systems into more accessible, knowledge-friendly formats through adaptive data segmentation and connectivity methods [32]. In practice, when employees participate in online meetings, GenAI tools can automatically generate meeting minutes, extract summaries from video or audio recordings, and organize key content—applications widely used on platforms like ZOOM. Based on knowledge management theory, Zou found that GenAI tools help dismantle interdepartmental knowledge barriers, facilitating cross-disciplinary creative integration and enabling breakthrough solutions [34]. Mu further demonstrated that AI-assisted brainstorming tools significantly enhance team innovation by generating diverse creative solutions [35]. Therefore, the following hypothesis is proposed:

 H1: Cognitive use of GenAI tools positively influences employees' knowledge transfer.

### 3.2. Social use of GenAI tools and resource acquisition capabilities

Resource acquisition is essential for workers, encompassing access to information, materials, office space, and sufficient working time. Within an organization, the acquisition of valuable resources comes from both internal sources and external entities such as customers, suppliers, competitors, universities, government agencies, and technology intermediaries [36]. The introduction of GenAI tools facilitates the creation of informal relationship networks, aligning with social exchange theory, by connecting individuals with shared interests and backgrounds, thereby expanding users' social networks [37,38]. Organizational members are encouraged to promote collaborative tools, social sharing platforms, and interactive communication channels, which foster team connections, enhance productive teamwork, and cultivate a sense of belonging among employees. When used for social motivation, GenAI tools enable employees to access crucial knowledge resources, alleviating the pressure on their personal resource usage. Noy and Zhang demonstrated that ChatGPT

users achieved increased writing efficiency while significantly reducing their time spent at work [39]. Qin observed that the implementation of AI technology improves work distribution and boosts employee creativity by alleviating human resource constraints [40]. Zhan and Li emphasized that the informational advantage in a generative environment greatly enhances cognitive resource efficiency and situational awareness [41]. Through GenAI tools, employees can more easily communicate, share information within and outside the organization, and strengthen emotional connections with both internal and external stakeholders. These social networks—comprising various internal and external members of the enterprise—serve as a foundation of trust, supporting employees' resource acquisition efforts and enhancing their ability to secure resources [42,43]. Based on these insights, the following hypothesis is proposed:

H2: Social use of GenAI tools positively influences employees' resource acquisition capability.

### 3.3. Knowledge transfer, resource acquisition, and job satisfaction

As previously discussed, knowledge transfer involves the sharing of knowledge across organizational boundaries or between individuals, encompassing the dissemination of various forms of knowledge within or across different entities [44]. Resource acquisition ability refers to employees' capacity to access relevant resources that facilitate their job tasks [31]. Employee job satisfaction is a multifaceted evaluation of work, representing the positive emotional state derived from assessing job experiences [45].

Knowledge transfer plays a pivotal role in knowledge management and is linked to enhanced employee job satisfaction [46–49]. By encouraging employees to voluntarily share their experiences, skills, and job-related knowledge, knowledge transfer improves work efficiency [48,49]. For example, Usmanova found that knowledge-sharing behaviors among skilled IT employees significantly contributed to their job satisfaction [48]. Furthermore, cultivating a corporate culture that promotes knowledge transfer fosters a supportive interpersonal environment, strengthening work relationships and enhancing job satisfaction [50–52]. Research also indicates that knowledge sharing and transfer generate opportunities for concept development and acquisition of job-related information, which leads to greater job satisfaction [51,53]. As knowledge transfer practices become more accessible in organizational settings, job satisfaction increases, as evidenced by Varshney and Damanhouri [54]. Based on these findings, the following hypothesis is proposed:

H3: Knowledge transfer positively influences employees' job satisfaction.

Previous studies have established a positive relationship between resource acquisition capabilities and job satisfaction [55,56]. Resource acquisition ability is reflected in employees' capacity to solve work-related problems, such as accessing external tacit knowledge, acquiring additional resources to complete tasks, and similar factors [31]. A higher level of resource acquisition ability suggests that employees are more likely to achieve a productive work state, maintain focus, and better assimilate organizational goals, ultimately leading to increased job satisfaction [56–58]. When employees adopt GenAI tools with an openness to change and innovation, coupled with a positive attitude, they can foster effective collaboration, strengthen their sense of identity and well-being, and improve job satisfaction [59–62]. For example, Wang found that GenAI's automated knowledge integration function reduces information overload, enabling employees to focus more on core tasks and thereby enhancing job satisfaction [61]. Gayathri and Bella highlighted that GenAI improves employees' ability to quickly access internal and external resources [62]. When applied to social media platforms, GenAI supports content generation or enhances user experiences, boosting employees' familiarity with their work, improving task completion abilities, and fostering more positive work attitudes, all contributing to greater job satisfaction. Based on these insights, the following hypothesis is proposed:

H4: Resource acquisition ability positively influences the improvement of job satisfaction.

### 3.4. Job satisfaction and employees' innovative job performance

IJP refers to employees' activities that exceed standard expectations, including the generation of creative ideas, problem-solving approaches, and the identification of new solutions [5,63,64]. For organizations to thrive, enhancing

competitiveness is essential, and this competitiveness is driven by innovation. Therefore, employees' IJP is a key factor in strengthening organizational competitiveness [65,66].

Extensive research suggests that high job satisfaction directly enhances innovative performance. First, when employees are satisfied with their jobs, they tend to enjoy their work more and demonstrate a greater willingness to perform [67]. Schappe found that higher job satisfaction leads to more positive perceptions of the organization, improved work performance, and a stronger inclination to engage in behaviors that benefit the organization [68]. Xiao observed that highly satisfied employees are more likely to internalize organizational goals, which, in turn, enhances their innovative performance through greater organizational commitment [69]. Second, job satisfaction is closely linked to the organizational innovation climate, which positively influences employees' innovative performance [43,70,71]. Yan and Bai found that high-performance work systems promote a positive innovation climate, boosting job satisfaction and fostering innovative behaviors [72]. Finally, job satisfaction influences employees' IJP by shaping their emotional state, which in turn affects motivation and creativity [73–75]. Amabile argued that positive emotional states, such as job satisfaction, enhance cognitive flexibility and the willingness to take risks, thereby boosting innovative output [73]. Zhou and George suggested that job satisfaction indirectly encourages innovative behaviors by increasing intrinsic motivation, such as a deeper interest in the task itself [75]. Based on these findings, the following hypothesis is proposed:

H5: Job satisfaction positively influences the improvement of employees' innovative job performance.

The primary research model of this study is depicted in Fig 1.

## 4. Research design

### 4.1. Data collection and sample selection

**4.1.1. Questionnaire recipients.** This study employed a comprehensive sampling strategy, distributing surveys *via* the Credamo platform to office workers across China. The aim was to identify general patterns in the relationship between GenAI usage and innovative performance, rather than examining differences across specific organizational types. This approach enhances the reliability and generalizability of the findings to common workplace contexts. This study involving human subjects was reviewed and approved by the Ethics Committee of Anhui Normal University (Approval No: AHNU-ET2025060). Written informed consent was obtained from all participants.

**4.1.2. Questionnaire design.** To minimize bias, the dependent variable scales were placed before the independent variable scales in the survey, ensuring that participants were not influenced by the independent variable when responding to dependent variable questions. Response quality was further ensured by setting a time requirement in the instruction section, allowing participants adequate time to review the instructions before completing the questions.

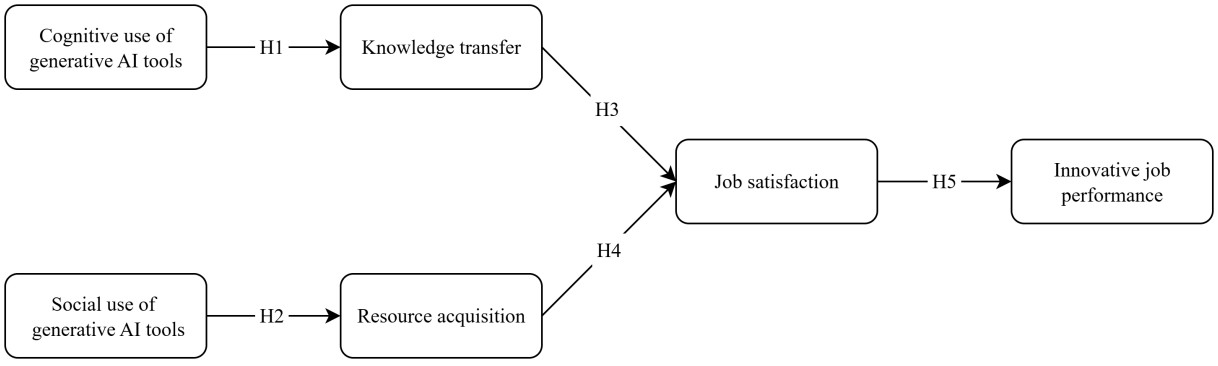

**Fig 1. Theoretical framework.**

An explanation of GenAI tool applications in the workplace was provided at the beginning of the questionnaire, along with screening questions to verify participants' professional experience. The survey was exclusively distributed through Credamo, with participants accessing it digitally *via* mobile devices or personal computers. (Specific questionnaire content is included in the Appendix).

**4.1.3. Questionnaire distribution and collection.** Participants were informed at the start of the questionnaire about the study's focus on their behaviors, attitudes, and work conditions. A confidentiality statement assured them that the collected data would be used solely for academic purposes, with explicit assurances that no information would be shared with employers or third parties. To incentivize participation and ensure response authenticity, participants received a CNY 2 payment for completing the survey.

In February 2025, 400 questionnaires were distributed *via* the Credamo platform. After removing 34 invalid responses following an attention check, 366 valid responses remained, resulting in a response validity rate of 91.5%.

**4.1.4. Analysis of demographic information.** The questionnaire collected demographic data such as age, gender, educational background, and marital status. Notably, 68.9% of the respondents were female, 57.4% were aged between 26 and 35, and over 90% of participants held a bachelor's degree or higher (Table 1).

## 4.2. Variable measurement

The measurement items for this research were adapted from established English scales in academic literature. To ensure linguistic accuracy, a back-translation process was conducted by bilingual marketing doctoral candidates, followed by multiple verification stages. A preliminary study with 30 participants was carried out to test and refine the wording and expressions before the data collection phase (CR = 0.749–0.958; AVE = 0.501–0.822). All constructs were measured using a 7-point Likert scale, ranging from 1 (strongly disagree) to 7 (strongly agree).

**4.2.1. Cognitive use of GenAI tools.** Cognitive use of GenAI tools was measured using a modified version of Ali-Hassan et al.'s social media usage scale [17]. Three items were included: "I use generative AI software (such as Doubao, KIMI, ChatGPT, DeepSeek, etc.) to collect work-related information and share content," "I use generative AI software... to disseminate content at work," and "I access content created by others using generative AI software..."

**4.2.2. Social use of GenAI tools.** Social use of GenAI tools was assessed using adapted items from Ali-Hassan et al.'s framework [17]. This construct was measured with five items, such as: "I use generative AI software... to build new relationships at work," "I use generative AI software... to connect with people I wouldn't meet at work," and "I use generative AI software... to maintain close social connections with colleagues at work." (The full set of items is provided in the Appendix.)

**Table 1. Demographic analysis.**

| Demographic variables | | Sample size | Percentage |
|---|---|---|---|
| **Gender** | Male | 114 | 31.1% |
| | Female | 252 | 68.9% |
| **Age** | Under 25 | 96 | 26.2% |
| | 26-35 | 210 | 57.4% |
| | 36 and above | 60 | 16.4% |
| **Education** | High school and below | 8 | 2.2% |
| | Junior college | 16 | 4.4% |
| | Undergraduate degree | 266 | 72.7% |
| | Master degree or above | 76 | 20.8% |
| **Marital status** | Married | 213 | 58.2% |
| | Spinsterhood | 153 | 41.8% |

**4.2.3. Knowledge transfer.** Knowledge transfer was evaluated using the scale developed by Cao et al [76]. Key items included: "I acquired technical written knowledge through GenAI tools (e.g., Doubao, KIMI, ChatGPT, DeepSeek)," "I learned management techniques *via* GenAI tools," "I gained new work experience through GenAI tools," and "I obtained knowledge about corporate culture from GenAI tools."

**4.2.4. Resource acquisition.** Resource acquisition was measured using an adapted version of Spreitzer's resource acquisition scale [31], with items such as: "I can access necessary resources to support innovative ideas in my work," "I typically obtain additional resources when required for task completion," and "I secure essential resources to perform my job effectively."

**4.2.5. Job satisfaction.** Job satisfaction was measured using Robertson and Kee's job satisfaction scale [77], which included items like: "I feel reasonably satisfied with my current job," "I maintain enthusiasm for my work most of the time," and "I genuinely enjoy performing my job duties."

**4.2.6. Innovative work performance.** Innovative work performance was assessed using Janssen's scale [78], with items such as: "Generate novel ideas for process improvement," "Mobilize support for innovative concepts," and "Explore original approaches to work tasks".

## 5. Result analysis

### 5.1. Reliability and validity test

TmartPLS 4.0 was used to analyze both the measurement and structural models, with the results summarized in Tables 2 and 3. The evaluation of the measurement model included internal consistency reliability, convergent validity, and discriminant validity. Internal consistency was assessed using composite reliability (CR), as recommended by Fornell and Larcker [79]. Convergent validity was evaluated through average variance extracted (AVE) and indicator loadings. Discriminant validity was determined by comparing the square root of the AVE to the inter-construct correlation coefficients.

As shown in Table 2, the six latent variables—cognitive use of GenAI tools, social use of GenAI tools, knowledge transfer, resource acquisition, job satisfaction, and innovative work performance—achieved CR values ranging from 0.777 to 0.953, all exceeding the 0.7 threshold, indicating strong internal consistency. In terms of convergent validity, factor loadings for all latent variables in Table 2 were above 0.6, with a minimum of 0.621 and a maximum of 0.920. The AVE values for all latent factors were greater than 0.5, ranging from 0.528 to 0.801, confirming high convergent validity, as per Fornell and Larcker [79]. For discriminant validity, the Fornell-Larcker Criterion was applied. The square root of the AVE for each latent factor exceeded the correlation coefficient between that factor and any other latent factor (Table 3), affirming the model's robust discriminant validity.

### 5.2. Structural equation model analysis

The first step in assessing structural modeling involves examining potential covariance issues among the latent variables. In PLS-SEM, covariance is typically assessed using the variance inflation factor (VIF). For the two predictor variables of satisfaction—knowledge transfer and resource acquisition—the VIF was 1.699, well below the critical threshold of 5, indicating no covariance issues among the predictor variables in the structural model. The significance of the path relationships in the structural equation model was evaluated using the Bias-Corrected and Accelerated (BCa) Bootstrap method with a two-tailed test at a 0.05 significance level. The procedure converged after three iterations (Fig 2, Table 4).

As presented in Table 4, the cognitive use of GenAI tools ($\beta = 0.689$, $p < 0.001$) significantly and positively influenced employee knowledge transfer, with the 95% confidence interval excluding zero, thereby supporting Hypothesis H1. Likewise, the social use of GenAI tools ($\beta = 0.558$, $p < 0.001$) significantly enhanced employee resource acquisition, with the 95% confidence interval also excluding zero, confirming Hypothesis H2. Additionally, employee knowledge transfer was found to have a significant positive impact on job satisfaction ($\beta = 0.361$, $p < 0.001$), with the 95% confidence interval

**Table 2. Model aggregation validity analysis table.**

| Variable | Measurement item | Factor loading value | Composite Reliability | AVE value |
|---|---|---|---|---|
| **Cognitive use of GenAI tools** | CU 1 | 0.709 | 0.819 | 0.602 |
| | CU 2 | 0.820 | | |
| | CU 3 | 0.794 | | |
| **Social use of GenAI tools** | SU 1 | 0.875 | 0.953 | 0.801 |
| | SU 2 | 0.884 | | |
| | SU 3 | 0.886 | | |
| | SU 4 | 0.920 | | |
| | SU 5 | 0.910 | | |
| **Knowledge transfer** | know_tra1 | 0.621 | 0.816 | 0.528 |
| | know_tra2 | 0.794 | | |
| | know_tra3 | 0.711 | | |
| | know_tra4 | 0.768 | | |
| **Resource acquisition** | Acq_res1 | 0.825 | 0.858 | 0.669 |
| | Acq_res2 | 0.811 | | |
| | Acq_res3 | 0.817 | | |
| **Job satisfaction** | Job_Stf1 | 0.801 | 0.934 | 0.738 |
| | Job_Stf2 | 0.810 | | |
| | Job_Stf3 | 0.894 | | |
| | Job_Stf4 | 0.886 | | |
| | Job_Stf5 | 0.898 | | |
| **Innovative work performance** | I_Job_Pf1 | 0.784 | 0.777 | 0.539 |
| | I_Job_Pf2 | 0.741 | | |
| | I_Job_Pf3 | 0.673 | | |

All factor loading coefficients are significant at the 1% significance level.

**Table 3. The mean, standard deviation, and correlation coefficient of the variable.**

| Variable | M | SD | CU | SU | know_tra | Acq_res | Job_Stf | I_Job_P |
|---|---|---|---|---|---|---|---|---|
| **CU** | 5.264 | 0.891 | 0.776 | | | | | |
| **SU** | 4.539 | 1.472 | .608*** | 0.895 | | | | |
| **know_tra** | 5.316 | 0.822 | .689*** | .646*** | 0.727 | | | |
| **Acq_res** | 5.313 | 0.874 | .600*** | .558*** | .641*** | 0.818 | | |
| **Job_Stf** | 5.090 | 1.134 | .611*** | .607*** | .670*** | .713*** | 0.859 | |
| **I_Job_P** | 5.659 | 0.701 | .568*** | .420*** | .578*** | .532*** | .556*** | 0.734 |

* $p < 0.05$, ** $p < 0.01$, *** $p < 0.001$; The value on the diagonal represents the square root of the AVE value of each latent variable.

excluding zero, supporting Hypothesis H3. Employee resource acquisition also significantly contributed to job satisfaction ($\beta = 0.481$, $p < 0.001$), with the 95% confidence interval excluding zero, validating Hypothesis H4. These findings suggest that employees' cognitive use of GenAI tools facilitates knowledge transfer, thereby enhancing job satisfaction, while their social use of GenAI tools aids in resource acquisition, which also increases job satisfaction. Finally, job satisfaction significantly enhanced innovative work performance ($\beta = 0.556$, $p < 0.001$), confirming Hypothesis H5.

Regarding variance explanation, the R Square value for knowledge transfer was 0.475 ($p < 0.001$), indicating that the cognitive use of GenAI tools explains 47.5% of the variance in knowledge transfer. The R Square value for

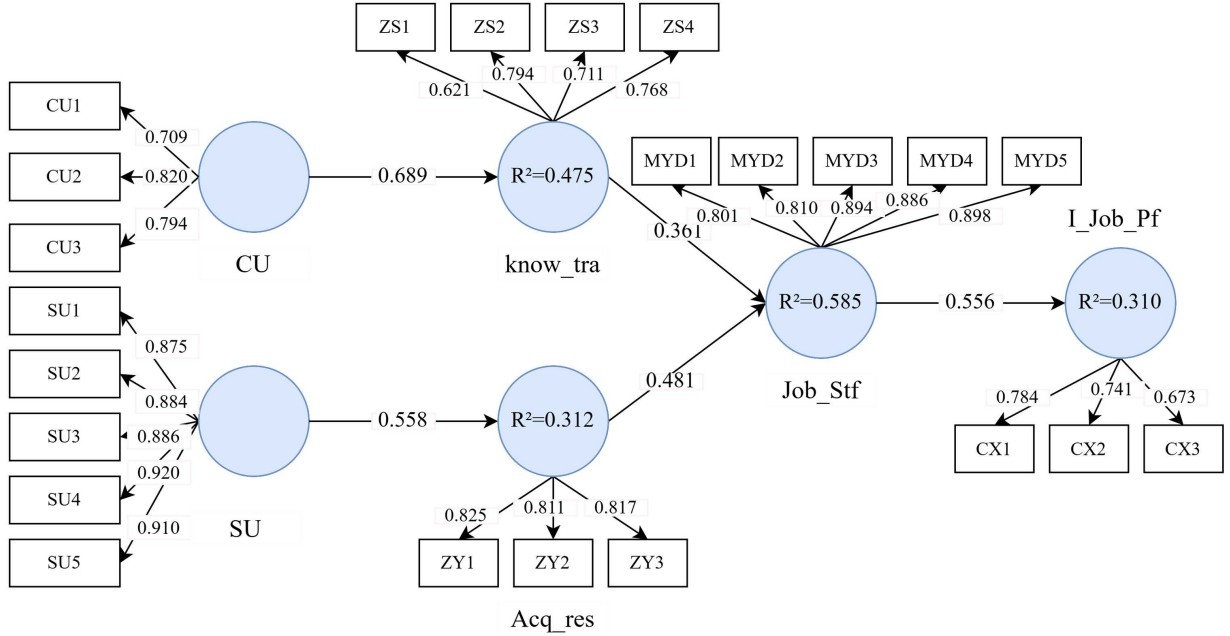

**Fig 2. Results of structural equation modeling.**

**Table 4. Path Coefficients and Significant Levels.**

| Path | β | t | 95% confidence interval | p | Hypothesis | f² | Effect size |
|---|---|---|---|---|---|---|---|
| **CU→know_tra** | 0.689 | 23.726 | [0.631, 0.746] | 0.000 | H1 | 0.906 | Large |
| **SU→Acq_res** | 0.558 | 13.093 | [0.473, 0.639] | 0.000 | H2 | 0.453 | Large |
| **know_tra→Job_Stf** | 0.361 | 6.760 | [0.259,0.470] | 0.000 | H3 | 0.185 | Medium |
| **Acq_res→Job_Stf** | 0.481 | 9.612 | [0.380, 0.576] | 0.000 | H4 | 0.328 | Medium |
| **Job_Stf→I_Job_P** | 0.556 | 14.421 | [0.483, 0.632] | 0.000 | H5 | 0.448 | Large |

CU = Cognitive Use of GenAI Tools; SU = Social Use of GenAI Tools; Know_tra = Knowledge Transfer; Acq_res = Resource Acquisition; Job_Stf = Job Satisfaction; I_Job_Pf = Innovative Job Performance.

resource acquisition was 0.312 (p < 0.001), suggesting that the social use of GenAI tools accounts for 31.2% of the variance in resource acquisition. The R Square value for job satisfaction was 0.585 (p < 0.001), meaning that knowledge transfer and resource acquisition together explain 58.5% of the variance in job satisfaction. Finally, the R Square value for IJP was 0.310 (p < 0.001), indicating that job satisfaction explains 31% of the variance in innovative work performance (Fig 2).

The effect size of each latent variable on the dependent variables was further assessed using $f^2$ analysis, complementing the $R^2$ analysis [80]. According to Cohen's criteria, $f^2$ values between 0.02 and 0.15 indicate small effects, between 0.15 and 0.35 indicate medium effects, and values above 0.35 suggest large effects [81]. Table 4 presents the $f^2$ results, showing three relationships with large effects and two with small effects. The findings demonstrate that the cognitive use of GenAI tools has a strong predictive effect on knowledge transfer, the social use of GenAI tools strongly predicts resource acquisition, and job satisfaction strongly predicts IJP. Additionally, the $f^2$ values for knowledge transfer (0.185) and resource acquisition (0.328) on job satisfaction indicate medium-strength predictive power for job satisfaction.

To assess the explanatory power of the predictors—knowledge transfer, resource acquisition, job satisfaction, and IJP variance ($R^2$)—this study also used the Blindfolding algorithm in SmartPLS 4.0 to calculate the $Q^2$ value. With a sample size of 366 and an omission distance of 7, the results indicated that the $Q^2$ values for repurchase intention, recommendation intention, and audience satisfaction were all greater than the critical threshold of 0 [82], confirming strong predictive relevance (Table 5).

### 5.3. Mediation effect analysis

To examine whether knowledge transfer, resource acquisition, and job satisfaction mediate the relationship between the use of AI tools for both cognitive and social purposes and IJP, PLS-SEM was employed to analyze the indirect, direct, and total effects among the variables, following the latest research and recommendations from scholars worldwide [82,83]. The mediating effects were analyzed using the Bootstrapping method in PLS-SEM, with the mediating effects categorized into five types (Table 6).

The complete chained mediation paths in this study reveal two distinct types. First, for the cognitive use of GenAI tools (CU), a complementary mediation is observed. Cognitive use has a significant direct effect on innovation performance ($\beta = 0.375$, $p < 0.001$) as well as a significant indirect effect through knowledge transfer and job satisfaction ($\beta = 0.082$, $p < 0.001$). This suggests that cognitive use partially influences innovation performance through mediating mechanisms while also exerting an independent direct effect. Second, for the social use of GenAI tools (SU), an indirect-only mediation is found. The direct effect of social use on innovation performance is not significant ($\beta = -0.006$, $p = 0.928$), but the indirect effect through resource acquisition and job satisfaction is significant ($\beta = 0.088$, $p < 0.001$). This indicates that social use influences innovation performance entirely through mediating mechanisms(Table 7).

## 6. General discussion

This study utilized UGT to examine how employees' use of GenAI tools, driven by different motivations, influences their IJP. The analysis revealed that both cognitively and socially motivated use of GenAI tools significantly enhances IJP. Specifically, cognitively motivated use of GenAI tools promotes knowledge transfer behaviors. For example, employees' cognitive use of these tools optimizes knowledge management, reduces their workload, and increases the efficiency of knowledge transfer. This not only strengthens employees' professional capabilities but also establishes a solid knowledge base for innovative behaviors. In contrast, socially motivated use of GenAI tools significantly enhances employees'

Table 5. The predictive quality of the Model.

| Total | SSO | SSE | $Q^2$=1-SSE/SSO |
|---|---|---|---|
| **Knowledge transfer** | 1464.000 | 1104.373 | 0.246 |
| **Resource acquisition** | 1098.000 | 873.514 | 0.204 |
| **Job satisfaction** | 1830.000 | 1049.169 | 0.427 |
| **Innovative work performance** | 1098.000 | 922.370 | 0.160 |

Table 6. Types and criteria of mediating effect.

| Serial number | Typology | Criterion |
|---|---|---|
| 1 | No-effect nonmediation | Neither direct nor indirect effects are significant |
| 2 | Direct-only nonmediation | Significant direct effect, but not significant indirect effect |
| 3 | Complementary mediation | Both direct and indirect effects are significant and in the same direction |
| 4 | Competitive mediation | Both direct and indirect effects are significant, but not in the same direction |
| 5 | Indirect-only mediation | Indirect effects are significant, but direct effects are not significant |

**Table 7. Mediator analysis.**

| Hypotheses | Indirect effects | t | 95% CI | p | Type of Intermediary |
|---|---|---|---|---|---|
| Cu→Know_tra→Job_Stf→I_Job_Pf | 0.082 | 3.556 | [0.037, 0.127] | 0.000 | Complementary mediation |
| Cu→Know_tra→Job_Stf | 0.249 | 6.454 | [0.172, 0.326] | 0.000 | / |
| Know_tra→Job_Stf→I_Job_Pf | 0.119 | 3.606 | [0.054,0.184] | 0.000 | / |
| Su→Acq_res→Job_Stf→I_Job_Pf | 0.088 | 4.153 | [0.047, 0.129] | 0.000 | Indirect-only mediation |
| Su→Acq_res→Job_Stf | 0.269 | 7.151 | [0.194, 0.344] | 0.000 | / |
| Acq_res→Job_Stf→I_Job_Pf | 0.158 | 4.384 | [0.087, 0.229] | 0.000 | / |

This table only determines the type of mediation for complete chained mediation paths, as their direct effects (CU→CX and SU→CX) have been tested in the supplementary analysis. The mediation type of certain paths (e.g., CU→Know_tra→Job_Stf) is not determined because their corresponding direct effects (e.g., CU→Job_Stf) are not included as direct paths in the theoretical model, and thus the classification criteria in Table 6 cannot be applied.

knowledge acquisition abilities. For instance, employees can leverage these tools to share experiences, discuss challenges, or seek assistance from colleagues, fostering cross-departmental collaboration and facilitating access to additional work resources. These cognitive and social pathways both contribute to increased job satisfaction, which in turn boosts IJP. Specifically, the use of GenAI tools driven by cognitive and social motivations indirectly enhances job satisfaction by improving knowledge transfer and knowledge acquisition. Efficient knowledge transfer enables employees to acquire new skills and knowledge more quickly, resulting in a sense of achievement and satisfaction with their work. Enhanced access to resources, on the other hand, provides employees with more opportunities for development and growth, boosting their confidence and optimism about career prospects. These positive work experiences collectively enhance job satisfaction. Satisfied employees are more likely to invest greater effort and time into their work, demonstrating higher motivation and creativity, which subsequently improves innovative work performance.

## 6.1. Theoretical contributions

First, this study expands the scope of UGT, which has traditionally been applied primarily in communication studies. In recent years, its application has extended to areas such as employee entrepreneurship and social media [13,17,18]. This paper further extends the UGT framework by examining the use of GenAI tools in the workplace. It empirically investigates how cognitive and social motivations drive employees' use of these tools, enhancing their knowledge transfer and resource acquisition capabilities, which in turn affect their job satisfaction and IJP. This extension not only enriches the theoretical application of UGT but also offers a fresh perspective on understanding the mechanisms through which GenAI tools shape employee behavior.

Second, this study adds to the literature on the impact of GenAI tools on employee behavior. Previous research on the relationship between GenAI tool usage and employee performance has largely focused on isolated aspects of use, often neglecting the underlying motivations and the complex mechanisms through which these tools influence performance [84–87]. By emphasizing the motivations driving employees' use of technology, this paper identifies two primary categories: cognitive and social motivations. Cognitive motivation stems from intrinsic needs such as knowledge acquisition, problem-solving, and skill enhancement, while social motivation addresses extrinsic needs such as social interaction, relationship building, and a sense of belonging. This framework provides a comprehensive analysis of how these motivations mediate the effect of GenAI tool usage on employees' innovative work performance, offering new insights into the relationship between GenAI and employee innovation.

Third, this study confirms the internal mechanisms linking employee knowledge sharing, resource acquisition, and job satisfaction. It not only highlights the direct impact of GenAI tool usage on IJP but also explores the intricate connections between employees' knowledge-sharing and resource acquisition abilities and their job satisfaction. These factors serve

as key mediators in the relationship between GenAI tool use and innovative performance. Specifically, the cognitive and social motivations driving the use of generative AI tools indirectly enhance job satisfaction by improving knowledge transfer and acquisition capabilities, which in turn fosters IJP [48,49,59–62,64,74,75]. This theoretical framework not only clarifies how GenAI tools contribute to enhancing employee innovation performance but also offers a solid model for future research in this field.

### 6.2. Practical implications

The findings of this study provide valuable insights into whether organizations should impose restrictions on employee use of GenAI tools. On the positive side, the integration of GenAI tools within organizations proves highly effective. These tools help address challenges such as file transfers in the workplace and streamline daily tasks. Moreover, they enhance interpersonal communication, dismantle information silos, and foster collaboration among team members. Employees' use of GenAI tools also significantly impacts their job satisfaction, which in turn is closely linked to their performance outcomes. Given these findings, organizations should focus on creating a high-quality technological environment that fully engages employees and encourages the active use of these tools. The introduction of GenAI tools facilitates more efficient knowledge transfer and acquisition, granting employees easy access to a wealth of resources that can expand their thinking and enhance their innovation capabilities. Additionally, companies must continuously refine the functionality and user experience of these tools to better align with employees' needs, stimulating positive feedback and encouraging proactive engagement. This approach not only enhances individual performance but also drives synchronized growth in overall organizational performance, resulting in mutual benefits for both employees and the organization.

As GenAI tools become more prevalent in the workplace, they present both opportunities and challenges. Enterprise management must explore the principles and business value of effectively utilizing GenAI tools to drive corporate performance improvements and achieve organizational objectives. Simultaneously, employees should be actively guided on the correct and efficient use of these tools to foster a positive, innovative work culture. This approach ensures a win-win outcome for both the enterprise and its employees, fostering deep intellectual and emotional integration with their work—key factors for the enterprise's long-term success. The growing prevalence of GenAI tools is an irreversible trend, making a return to a pre-GenAI era impossible. Therefore, the focus should be on fully leveraging their potential to support organizational growth, enhancing employee productivity by utilizing platforms like social media to unlock the substantial potential of these tools in the workplace.

### 6.3. Limitations and future research

While this study confirms the positive impact of GenAI tools on employees' innovative job performance, several limitations remain, offering avenues for further exploration.

First, from a methodological perspective, the cross-sectional design limits the ability to establish causal inferences. Although the hypothesized relationships are grounded in theoretical reasoning and empirical support, the reliance on a single time point of data collection prevents definitive conclusions about the direction of causality. Future research could employ longitudinal or experimental designs to better capture the dynamic and causal nature of the relationships examined.

Second, the research primarily relied on self-reported survey data, which may introduce common-method bias. To enhance validity, future studies could incorporate multi-source data (e.g., supervisor ratings, objective performance indicators) or longitudinal approaches, and employ statistical techniques such as hierarchical or multi-level modeling to mitigate potential biases.

Third, the sample primarily reflects general enterprise employees, 90% of whom hold at least a bachelor's degree. This limits the generalizability of the findings across industries, job levels, and educational backgrounds. Future research could investigate whether the effects of GenAI vary across sectors such as healthcare, finance, or manufacturing, as

well as across different managerial hierarchies and regional contexts. Additionally, personal factors—such as educational attainment, openness to experience, and digital literacy—may moderate the influence of GenAI on innovation outcomes. Exploring these moderating effects could provide organizations with more targeted guidance for employee-focused AI strategies.

Fourth, while this study highlights the positive indirect impact of cognitive and social GenAI use on IJP, it does not address potential negative outcomes. Overreliance on GenAI tools may lead to issues such as information overload, reduced cognitive engagement, or "social saturation," where constant digital interaction hinders creativity. Future research should examine both the enabling and constraining effects of GenAI use to capture its dual influence on innovation more comprehensively.

Finally, the ethical considerations surrounding GenAI use in the workplace warrant further investigation. Issues such as data privacy, algorithmic bias, and transparency in AI-assisted decision-making raise important questions regarding fairness and accountability. Employees' perceptions of ethical AI governance may influence their trust in technology and their willingness to engage in innovation. Future research could explore how ethical AI management frameworks—such as transparent model explanations or privacy-by-design practices—shape employees' attitudes, psychological safety, and creative performance.

In conclusion, while this study provides empirical evidence of GenAI's beneficial role in fostering innovative job performance, future investigations should adopt a broader, ethically informed, and methodologically diverse approach to fully understand the complexities of AI-driven innovation in organizational contexts.

## Supporting information

**S1 File. Appendix 1 questionnaire.**
(DOCX)

**S1 Data. AI sample.**
(XLSX)

## Author contributions

**Conceptualization:** Hui Zhang, Lidong Zhu.

**Data curation:** Hui Zhang.

**Formal analysis:** Ayuan Zhang.

**Funding acquisition:** Hui Zhang.

**Resources:** Hui Zhang.

**Software:** Ayuan Zhang.

**Supervision:** Lidong Zhu, Kilichov Shohruh.

**Writing – original draft:** Hui Zhang.

**Writing – review & editing:** Lidong Zhu, Ayuan Zhang, Kilichov Shohruh.

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
