## [Decision Letter · Decision Letter 0]

21 Apr 2025

Dear Dr. Zhang,

Thank you for submitting your manuscript to PLOS ONE. After careful consideration, we feel that it has merit but does not fully meet PLOS ONE’s publication criteria as it currently stands. Therefore, we invite you to submit a revised version of the manuscript that addresses the points raised during the review process.

We look forward to receiving your revised manuscript.

Kind regards,

Uğur Cakilcioğlu, PhD

Academic Editor

PLOS ONE

2. You indicated that ethical approval was not necessary for your study. We understand that the framework for ethical oversight requirements for studies of this type may differ depending on the setting and we would appreciate some further clarification regarding your research. Could you please provide further details on why your study is exempt from the need for approval and confirmation from your institutional review board or research ethics committee (e.g., in the form of a letter or email correspondence) that ethics review was not necessary for this study? Please include a copy of the correspondence as an ""Other"" file.

“This research was funded by the National Natural Science Foundation of China (NSFC), grant number “72202002”.”

4. In the online submission form you indicate that your data is not available for proprietary reasons and have provided a contact point for accessing this data. Please note that your current contact point is a co-author on this manuscript. According to our Data Policy, the contact point must not be an author on the manuscript and must be an institutional contact, ideally not an individual. Please revise your data statement to a non-author institutional point of contact, such as a data access or ethics committee, and send this to us via return email. Please also include contact information for the third party organization, and please include the full citation of where the data can be found.

Additional Editor Comments:

Dear authors;

The paper should be edited according to the writing rules of the journal

Original manuscript. There are, however, major changes required.

Regards

Reviewers' comments:

Reviewer's Responses to Questions

**Comments to the Author**

1. Is the manuscript technically sound, and do the data support the conclusions?

Reviewer #1: Partly

Reviewer #2: Partly

2. Has the statistical analysis been performed appropriately and rigorously?

Reviewer #1: No

Reviewer #2: No

3. Have the authors made all data underlying the findings in their manuscript fully available?

Reviewer #1: No

Reviewer #2: Yes

4. Is the manuscript presented in an intelligible fashion and written in standard English?

Reviewer #1: Yes

Reviewer #2: No

Reviewer #1: The Uses and Gratifications Theory (UGT) has been addressed extensively. The classification of motivations within this theory is explained at great length; it is recommended that this section be restructured more concisely and directly connected to the research model. Additionally, the relationship between UGT’s media-based origins and the context of AI use should be discussed more clearly and critically.

The current model focuses solely on cognitive and social motivations. However, other important components of UGT, such as hedonic motivation, have been excluded. This creates a lack of full alignment with the theory. The model should be expanded with new hypotheses to address this gap.

Although the study is based on cross-sectional data, it makes causal claims. This is a significant methodological issue. The authors should either soften the causal language or propose longitudinal or experimental studies to support such claims.

The contribution to the literature section repeats many concepts. Does this study merely apply the Uses and Gratifications Theory, or does it extend it? The claim of theoretical contribution should be strengthened.

The translation and adaptation process of the scales is described in a very superficial manner. The use of GenAI tools can vary across cultural contexts, particularly in China. The appropriateness of the scales for the Chinese cultural context should be supported with stronger justification.

In the structural model, the direct effect of social use on innovative performance appears negative and insignificant. This result has been overlooked by the authors. It should be seriously considered, discussed, or the rationale for including this variable in the model should be re-evaluated.

90% of participants have an undergraduate or higher level of education. The implications of this for the generalizability of the results should be discussed, as this sample may not represent the broader Chinese workforce.

The limitations section at the end of the paper is overly general. The effects of issues such as the ethical use of GenAI tools, privacy concerns, and algorithmic bias should be taken into account.

Data availability is stated as “available upon request,” but PLOS ONE requires an open data policy. The authors should clearly specify under what conditions and on which platform the data will be shared.

The literature heavily relies on Western-centric references. More up-to-date local sources and case studies specific to the Chinese workforce should be included.into account.

Reviewer #2: First of all, I congratulate you on your work. However, your article needs improvement in some important areas:

Reporting of statistical analyses is incomplete. In the Results section, important indicators such as beta coefficients, p-values, R² are not given in full in some places. I did not see that indirect effects (mediation) were tested. The overall explanatory power of the model could be expanded.

Discussion section could be organized. The findings are only repeated and the linking and comparison with the literature is weak. This section could be expanded to discuss similarities or differences in findings with previous studies.

Ethical approval process and data confidentiality are not adequately explained. Ethics committee approval or exemption statement for the research should be clearly stated; otherwise, it may pose a problem in terms of publication ethics.

The written language is below academic standards. One gets the impression that a large part of the article was created with a google translate translation. This negatively affects conceptual clarity and quality of expression. A professional English academic editing is required to make it suitable for publication.

Practical contributions and recommendations should be developed and presented more clearly for practitioners. At the moment the suggestions are very general and can be expanded as suggestions.

**Do you want your identity to be public for this peer review?** For information about this choice, including consent withdrawal, please see our Privacy Policy

Reviewer #1: No

Reviewer #2: No

---

## [Author Response · Author response to Decision Letter 1]

19 May 2025

Many thanks to the reviewers and editors for their suggestions; we have changed all the problems in the paper

---

## [Decision Letter · Decision Letter 1]

7 Sep 2025

Dear Dr. Zhang,

Thank you for submitting your manuscript to PLOS ONE. After careful consideration, we feel that it has merit but does not fully meet PLOS ONE’s publication criteria as it currently stands. Therefore, we invite you to submit a revised version of the manuscript that addresses the points raised during the review process.

I secured the opinion of one reviewer, and I have carefully read the manuscript myself to offer my independent opinion. Both the reviewer and I believe that there are some merits in the study, and it may be of interest to readers of PLOS One upon major revisions. The authors should carefully consider each of my comments and the Reviewer’s comments.

In the Introduction section, it is not clear to me what is meant by “social use of AI”? Adding some examples would be useful, together with the use of example items in the Method section (see below). I have a similar concern regarding “resource acquisition” that could be better explained in the Introduction section. In the scale used, the resource acquisition is not explained to participants, with possible different interpretations by them. In general, since the topic is very new and in development, I think that the Introduction can be updated with more recent work (even published in 2025).

In the Method section, the authors state that they target “employee groups”; what groups were targeted? How were the organizations selected? The type of organizations the participants work in should be included in the demographics.

“An explanation of workplace applications for GenAI tools was provided both at the start of the questionnaire and through screening questions to confirm participants' professional experience”. The authors should disclose this information and may include it in a Supplementary file.

In the descriptions of the scales, examples of items for cognitive and social use of AI are missing. Moreover, the response scales and range of values are missing for all instruments.

In the Results section, in Figure 2, a note should be added explaining the abbreviations used.

Moreover, the information in Table 6 is not justified since the authors do not use these classifications in their results. Regarding the model’s results, they should be better presented. For instance, it seems that Figure 2 shows only total effects. Where can the reader find direct effects?

A very important point is that the writing is very poor and should be improved. In some parts of the manuscript, it is difficult to understand what is meant, especially in the Introduction and Discussion sections. I encourage authors to revise the manuscript with the aim of enhancing clarification.

Regarding the response to Reviewers, I have similar concerns; it was difficult for me to follow. For instance, in the Response to Reviewer 1 point 3, the authors’ citation is not consistent with what was then reported in the manuscript (in blue).

All these issues should be carefully addressed for consideration of publication.

We look forward to receiving your revised manuscript.

Kind regards,

Annalisa Theodorou

Academic Editor

PLOS ONE

Journal Requirements:

Reviewers' comments:

Reviewer's Responses to Questions

**Comments to the Author**

Reviewer #2: All comments have been addressed

Reviewer #3: All comments have been addressed

Reviewer #4: All comments have been addressed

2. Is the manuscript technically sound, and do the data support the conclusions?

Reviewer #2: Yes

Reviewer #3: Yes

Reviewer #4: Yes

3. Has the statistical analysis been performed appropriately and rigorously?

Reviewer #2: Yes

Reviewer #3: Yes

Reviewer #4: Yes

4. Have the authors made all data underlying the findings in their manuscript fully available?

Reviewer #2: Yes

Reviewer #3: Yes

Reviewer #4: Yes

5. Is the manuscript presented in an intelligible fashion and written in standard English?

Reviewer #2: Yes

Reviewer #3: Yes

Reviewer #4: Yes

Reviewer #2: Dear Authors,

Thank you for submitting the revised version of your manuscript. Your study presents a relevant and timely investigation into the factors influencing the adoption of AI-based technologies in healthcare settings in Saudi Arabia. The integration of TOE and UTAUT frameworks strengthens the theoretical foundation and helps capture both organizational and individual-level factors.

Your use of a mixed-methods design is well justified and effectively implemented. The quantitative data is analyzed appropriately through descriptive and regression analyses, while the qualitative interviews complement and reinforce the findings. The conclusions are supported by the data, and you have clearly acknowledged the limitations of your study.

The manuscript is generally well-structured and intelligible, although we recommend a professional language edit to address minor grammatical and stylistic issues, particularly in the discussion and conclusion sections.

Overall, this is a technically sound and methodologically robust contribution that addresses a significant research gap in digital health transformation, particularly in the context of developing economies.

Congratulations on your work. I wish you continued success in your future research endeavors.

Best regards,

Reviewer #3: Dear Editor; The attached article was checked. The manuscript contains interesting information about The Influence of Using Generative Artificial Intelligence Tools on Employee's Innovative Job Performance

I think that this article is well suited to your journal.

It is generally good work. The scientific and presentation level of the manuscript is high.

The title is understandable and in line with the text. The text is written in a descriptive and understandable language. The material and method are well described and adequately detailed Discussion and conclusion are interrelated.

Reviewer #4: A few of the sentences in the intro and discussion are clumsy or wordy and could be helped by some light language editing for concision and clarity.

Figure 1 (conceptual model) could be redrawn for easier visual clarity. For example, use the same shapes and even spacing to make it easier to read.

You could mention ethical implications of GenAI use in the workplace (e.g., privacy, bias, algorithmic transparency) briefly, even speculatively.

**Do you want your identity to be public for this peer review?** For information about this choice, including consent withdrawal, please see our Privacy Policy

Reviewer #2: No

Reviewer #3: No

Reviewer #4: **Yes:**  Stefanos Balaskas

---

## [Author Response · Author response to Decision Letter 2]

1 Nov 2025

Dear Editor and Reviewers,

We sincerely appreciate the time and effort you devoted to reviewing our manuscript entitled “The Influence of Using Generative Artificial Intelligence Tools on Employee's Innovative Job Performance” (Manuscript ID: PONE-D-25-16015). We are truly grateful for your insightful comments and constructive suggestions, which have been invaluable in improving the clarity, rigor, and overall quality of our paper.

We have carefully revised the manuscript in accordance with all the comments provided by the editor and reviewers. Each suggestion has been thoughtfully considered, and corresponding changes have been made to strengthen the theoretical framework, refine the methodology, enhance the language, and enrich the discussion. In the following sections, we provide a detailed, point-by-point response explaining how each comment has been addressed in the revised version.

We deeply appreciate your constructive feedback and the opportunity to revise our work. Your guidance has greatly helped us enhance the scholarly contribution and readability of the manuscript.

---

## [Editor Report · Decision Letter 2]

20 Nov 2025

Dear Dr. Zhang,

Thank you for submitting your manuscript to PLOS ONE. After careful consideration, we feel that it has merit but does not fully meet PLOS ONE’s publication criteria as it currently stands. Therefore, we invite you to submit a revised version of the manuscript that addresses the points raised during the review process.

I carefully read the authors' responses to my comments and the reviewers' feedback. I believe that the authors did a good job in responding to each point. For this reason, I consider the manuscript accepted upon minor revisions. In particular, I kindly request that the authors submit the Appendix file, which is currently unavailable on the platform. Moreover, regarding response to Reviewer 1, Comment 3, I believe that in the modified insert in the manuscript, the reference to the important limitation regarding the cross-sectional nature of the data and the impossibility of establishing causal inference is now missing. I believe that this important limitation should be integrated before publishing the manuscript.

We look forward to receiving your revised manuscript.

Kind regards,

Annalisa Theodorou

Academic Editor

PLOS ONE
---

## [Author Response · Author response to Decision Letter 3]

6 Dec 2025

Dear Editor ,

We sincerely appreciate the time and effort you devoted to reviewing our manuscript entitled “The Influence of Using Generative Artificial Intelligence Tools on Employee's Innovative Job Performance” (Manuscript ID: PONE-D-25-16015). We are truly grateful for your insightful comments and constructive suggestions, which have been invaluable in improving the clarity, rigor, and overall quality of our paper.

We have carefully revised the manuscript in accordance with all the comments provided by the editor. Each suggestion has been thoughtfully considered, and corresponding changes have been made to strengthen the theoretical framework, refine the methodology, enhance the language, and enrich the discussion. In the following sections, we provide a detailed, point-by-point response explaining how each comment has been addressed in the revised version.

We deeply appreciate your constructive feedback and the opportunity to revise our work. Your guidance has greatly helped us enhance the scholarly contribution and readability of the manuscript.

1.We have now included the Appendix in the revised manuscript, placed after the References section. The Appendix contains the complete survey instrument used in this study, including all measurement items for the six constructs (cognitive use, social use, knowledge transfer, resource acquisition, job satisfaction, and innovative job performance). This addition enhances the transparency and reproducibility of our research.

2.We sincerely apologize for this oversight and thank you for highlighting this critical methodological point. We fully agree that the cross-sectional nature of our data represents a significant limitation for establishing causal directions. In direct response to your comment, we have thoroughly revised the “Limitations and Future Research” section (Section 6.3) to explicitly and prominently address this issue.

---

## [Editor Report · Decision Letter 3]

29 Dec 2025

The Influence of Generative Artificial Intelligence Usage on Employees’ Innovative Job Performance

PONE-D-25-16015R3

Dear Dr. Zhang,

We’re pleased to inform you that your manuscript has been judged scientifically suitable for publication and will be formally accepted for publication once it meets all outstanding technical requirements.

Kind regards,

Annalisa Theodorou

Academic Editor

PLOS One
---

## [Editor Report · Acceptance letter]

21 Apr 2025

PONE-D-25-16015R3

PLOS One

Dear Dr. Zhang,

I'm pleased to inform you that your manuscript has been deemed suitable for publication in PLOS One. Congratulations! Your manuscript is now being handed over to our production team.

Kind regards,

on behalf of

Dr. Annalisa Theodorou

Academic Editor

PLOS One